# Locomotion and Postural Control in Young Adults with Autism Spectrum Disorders: A Novel Kinesiological Assessment

**DOI:** 10.3390/jfmk9040185

**Published:** 2024-10-03

**Authors:** Riccardo Di Giminiani, Stefano La Greca, Stefano Marinelli, Margherita Attanasio, Francesco Masedu, Monica Mazza, Marco Valenti

**Affiliations:** Department of Biotechnological and Applied Clinical Sciences, University of L’Aquila, 67100 L’Aquila, Italy; stefano.lagreca@graduate.univaq.it (S.L.G.); stefano.marinelli@graduate.univaq.it (S.M.); margherita.attanasio@univaq.it (M.A.); francesco.masedu@univaq.it (F.M.); monica.mazza@univaq.it (M.M.); marco.valenti@univaq.it (M.V.)

**Keywords:** ASD, gait analysis, angle–angle diagram, balance, muscle activation, EMG activity

## Abstract

**Background/Objectives**: The purposes of the present study were to assess gait by using a novel approach that plots two adjacent joint angles and the postural control in individuals with autism (ASD) and individuals with typical neurodevelopmental (TD). **Methods**: The surface electromyography (sEMG) activity was measured synchronously with the other variables. Twenty young adult men, 10 with TD and 10 with a diagnosis of ASD, took part in this study. **Results:** There was a significant difference between ASD and TD groups in the area described by the knee–ankle diagram (*p* < 0.05). The sEMG activity recorded from the lateral gastrocnemius (LG) during the contact phase of gait was significantly lower in the ASD group compared with the TD group (*p* < 0.05). The sEMG activity recorded in the different postural conditions showed differences in LG and tibialis anterior (TA) between the ASD and TD groups (*p* < 0.05). **Conclusions**: The knee–ankle diagram provided a sensitive and specific movement descriptor to differentiate individuals with ASD from individuals with TD. The reduced LG activation is responsible for the reduced area in the knee–ankle diagram and ‘toe-walking’ in individuals with ASD and represents the common denominator of an altered ankle strategy during locomotion and postural control.

## 1. Introduction

Autism spectrum disorder (ASD) is a pervasive neurodevelopmental disorder. A recent estimate suggests that throughout the world, about 1 out of every 100 children is diagnosed with ASD [1]. This disorder is characterised by a wide variety of sensorimotor, behavioural, and/or physiological symptoms, including a deficit in social communication, narrowed interests, stereotype functions, and repetitive patterns of behaviour [2,3]. The neurodevelopmental alterations in individuals with ASD involve cortical and subcortical regions of white matter [4], including alterations in the corpus callosum, which connects the left and right hemispheres [5]. In addition, individuals with ASD present cerebellar abnormalities and/or atrophy, fewer cerebellar Purkinje cells, and differences in the relative size of the deep cerebellar nuclei. Alterations in the spinocerebellar tracts that innervate the cerebellum affect locomotion and postural control while standing upright and lead to other gross motor impairments. These alterations are congruent with deficits in feedforward control mechanisms and sensory feedback involving cerebellar circuits [2].

The cerebellum plays a prominent role in motor control and learning [6]. Deficits that result from abnormal cerebellar development could compromise the execution of many gross motor tasks and a range of cognitive, intellectual, linguistic, affective, and social functions [7,8,9], highlighting atypical development [10]. The association between cerebellar functioning and motor behaviour in individuals with ASD might be of interest to investigate the changes in the patterns of various motor tasks (locomotion, postural control, reaction time, and fine motor control) that in turn could be potentially diagnostic (bio)markers of neural alterations and provide accurate information underlying ASD [2]. Over the last decade, there has been increasing attention concerning the neuromotor profile in individuals with ASD. There is a need to understand the various aspects that characterise the movements of individuals with ASD and to clarify the neurodevelopmental variability in this population [2].

The neuromotor profile has often been assessed based on fundamental motor tasks, such as locomotion and postural control, given their functional significance in the activities of daily living. Several investigations have shown differences in locomotion and postural control in individuals with ASD compared with individuals with typical neurodevelopment (TD) [9,11,12,13,14,15]. According to Fournier et al. [11], the modifications in an individual’s neuromotor profile are not associated with their cognitive abilities. However, the neuromotor deficits during locomotion seem to be correlated with the severity of the ASD diagnosis [9], namely the ‘core deficits’ of ASD [16]. Weiss et al. [9] reported that individuals with more severe forms of ASD present the greatest differences in spatiotemporal parameters during gait. Bojanek et al. [17] and Travers et al. [18] reported an association between the severity of ASD symptoms and postural control. It is also interesting to note that the degree of severity of the ASD diagnosis has been related to the strength deficit [19]. Other studies suggest that postural control alterations in individuals with ASD are more pronounced compared with individuals with TD when the sensory integration demands are increased [17,20,21].

Regarding spatiotemporal variables, individuals with ASD tend to walk at slower speed, reducing their stride length and increasing their stance width [9,22], and they show a reduced lower extremity range of motion (ROM) compared with individuals with TD [22,23]. However, there have been conflicting results for the biomechanical variables. For example, some studies did not reveal differences between individuals with ASD and individuals with TD in the joint kinematic parameters (angle–time) [14,24], inter-limb asymmetry during gait [15,24], spatiotemporal variables during gait [12,15], and postural control [18,20,21]. The discrepancies could be partially explained by the considerable heterogeneity in sample sizes, ASD subtypes, age, gender, IQ, and different metrics used to assess the neuromotor profile [25,26,27]. Most of the available studies have involved young children (2–12 years old), while only a few have included young adults [9,23,28]. This age factor could explain the discrepancies of the published results, considering that the gait pattern develops during growth [29]. Based on the available literature, it is difficult to draw a definitive conclusion; thus, additional research including the standardised above cited variables and anthropometric matched controls is required.

Of note, the spatiotemporal variables and the relationships between joint angular changes over the stride cycle are dependent on the walking speed [30]. In other words, speed constraint is necessary, and it could be imposed by using a treadmill. To date, no investigation has controlled speed in the study design. In addition, these metrics do not always provide information that can be easily summarised and interpreted in clinical assessments over time as the typical common aspects of gait in individuals ASD involve several variables [9].

Recently, a different approach, based on the angle–angle diagram, has been used to assess locomotion. This approach provides individual qualitative (i.e., the shape) and quantitative information regarding ROM and coordination (i.e., area, perimeter) in people with multiple sclerosis [31]. The method plots the angle changes in adjacent joints to each other, thus forming a loop. The area delimited in this diagram quantifies the conjoint ROM (hip–knee or knee–ankle) performed during a gait cycle [30,31]. The perimeter of the loop quantifies the degree of coordination between two joints [32].

The angle–angle diagrams offer numerous advantages in relation to spatiotemporal variables. The data are presented in a manner that typically displays both temporal and angular data pertaining to two joints simultaneously, and distinct gaits exhibit easily identifiable characteristics as loops. The loops convey information regarding limitations in the ranges of motion, and the overall ranges of motion are readily appreciated due to the angle–angle diagrams’ position relative to a set of mutually perpendicular axes. Furthermore, events occurring on the contralateral limb can be reflected in angle–angle diagrams.

While there are no available angle–angle diagrams for individuals with ASD, this information could provide additional significant differences between individuals with ASD and individuals with TD during locomotion. The gait of individuals with ASD is characterised by altered kinematic parameters at the hip, knee, and ankle joint levels (i.e., ankle dorsiflexion in the swing phase and decreased peak plantarflexion and increased peak hip flexion angles in stance and the swing phase [22,23,33]). Hence, it is reasonable to expect typical common features in the loops defined by knee–ankle and hip–knee diagrams. Hip and ankle joints are also involved in the postural control strategies while standing upright [34,35,36,37]; therefore, altered neuromuscular patterns could be generated in the lower leg muscles in both motor tasks (i.e., locomotion and upright standing).

The purpose of the current study was to assess locomotion and postural control in young adult men with ASD compared with young adult men with TD. Specifically, we analysed gait based on the angle–angle diagrams and angle–time relationships obtained in a sagittal plane of motion at an imposed walking speed. We assessed postural control by analysing body sway while standing upright in dynamic and static conditions with open and closed eyes. We used surface electromyography (sEMG) to measure the activity of flexor and extensor muscles in the lower limbs during gait and body sway assessment. We hypothesised that compared with individuals with TD, individuals with ASD present (1) a reduced area of the angle–angle diagrams during walking, (2) increased body sway while standing upright, and (3) electromyographic differences in the lower leg muscles during walking and upright standing. Moreover, age, gender, and anthropometric characteristics (i.e., stature and body mass) can influence locomotion and upright standing and need to be defined. Information regarding locomotion and postural control while standing upright could provide relevant information that improves the daily safety of individuals with ASD, given that many people with high-functioning autism are university students or are engaged in work activities.

## 2. Materials and Methods

### 2.1. Participants and Experimental Procedure

Twenty young adult males, 10 with TD (age: 23.5 ± 2.5 years; height: 1.79 ± 9.7 m; body mass: 70.1 ± 10.9 kg; body mass index: 22.0 ± 2.6 kg/m^2^; leg preference: right leg) and 10 with ASD (age: 22.4 ± 3.4 years; height: 1.75 ± 5.3 m; body mass: 78.4 ± 10.6 kg; body mass index: 25.2 ± 3.9 kg/m^2^; leg preference: 9 right and 1 left) took part in this study. The sample size estimation for the primary outcomes (angle–angle diagram and sEMG synchronised during walking) was computed a priori by means of statistical software for power analysis (G*Power 3.1.9.4, Heinrich Heine-Dusseldorf University, Düsseldorf, Germany). The computation was performed in relation to the study design (*t*-test family for parametric and nonparametric distribution), setting the effect size (ES), and using the protocol for a power analysis (test attributes, large ES [1.8], α = 0.05, power [1 − β] = 0.96, total sample size n = 20 participants). The diagnosis of ASD was as follows: 8 with Level 1 and 2 with Level 2 (mean QI = 81.6, standard deviation = 18.8). The inclusion criteria were the absence of any history of injury, fracture, or morphological changes in the lower extremities. Individuals with ASD were recruited from the Reference Regional Centre for Autism (CRRA) in L’Aquila (Italy). They had been diagnosed according to the criteria of the Diagnostic and Statistical Manual of Mental Disorders (5th ed., DSM-5; American Psychiatric Association [APA], 2013) [38] and the Autism Diagnostic Observation Schedule-2 (ADOS-2) [39]. Each participant had ADOS-2 scores that were above the cut-off (mean communication and social interaction = 12.2 ± 2.8; mean stereotyped behaviours and restricted interests = 0.8 ± 0.8). The experimental protocol (0050370/21) was approved by the ethics committee of the NHS Local Health Unit (Azienda Sanitaria Locale 1), according to the ethical standards of the Declaration of Helsinki. The informed consent was obtained from all the participants and/or their legal guardians. The participants became familiar with the experimental procedure before the measurements (a different day). The measurements lasted about 2 h for each participant, and the protocol was carried out between 10:00 and 13:00 h. At the beginning of the assessment, each participant provided personal, anthropometric, and biometric data. Then, each participant underwent the gait and postural control assessment with synchronised sEMG.

### 2.2. sEMG Activity

sEMG was recorded with bipolar electrodes (Ambu Neuroline 720 00-S/25, electrode diameter 45 × 22, interelectrode distance 2.5 cm, Ambu A/S, Baltorpbakken, Denmark) in four dominant leg muscles: the vastus lateralis (VL), biceps femoris (BF), tibialis anterior (TA), and lateralis gastrocnemius (LG). The leg dominance was determined by using a standardised neuro-psychomotor assessment [40]. The electrodes were placed on the leg muscles according to the sEMG for non-invasive assessment of muscle recommendations [41]. Before application of the electrodes, the skin was shaved, slightly abraded with sandpaper, without producing redness, and cleaned with alcohol to minimise impedance (<5 kΩ). Prior to sensor placement, the exact location on the muscle belly was marked with indelible ink to ensure correct and standardised sensor positioning. The electrodes, the cables connected to the modules, and the modules were fixed with elastic strips to prevent motion artefacts. The wireless characteristics of the data synchronisation unit (Muscle-Lab Bosco-System 6000, Ergotest Innovation AS, Stathelle, Norway) were: a built-in ML6RFM02 radio frequency module (2.4 GHz and 1 mW) and a typical wireless range in open space of 20 m. The module characteristics were a radiofrequency of 2.4 GHz and 1 mW, a sample rate of 1 kHz, a high-pass filter of 20 Hz, a bandwidth of 20–500 Hz, an input signal range of ±5 mVp-p, noise of 14 µVp-p (2.2 µVrms), and a resolution of 0.33 µV/bit.

### 2.3. Gait Analysis and Synchronised sEMG

The kinematic gait analysis was performed by using the SMART Motion Capture System (BTS, Bioengineering, Milan, Italy), equipped with optoelectronic cameras able to reconstruct the three-dimensional (3D) trajectories provided by specific reflective markers placed on the anatomic joint points of the subjects. Four cameras were used to delimit the shooting volume around the treadmill, on which each participant walked for 30 s at 3 km/h while sEMG activity in the leg muscles was recorded. A 22-segment model was created for each subject (Figure 1A), and the reflective markers were placed at the seventh cervical vertebra, the first sacral vertebra, the acromion, the lateral epicondyle of the elbows, and the distal epiphysis of the radius of both upper limbs; in both lower limbs, the markers were placed at the femoral greater trochanter, the lateral condyle of the femur, the lateral malleolus, the calcaneus, and the distal epiphysis of the fifth metatarsal. The 9-element data were considered in the analysis of the lower limbs using the SMART Analyser (BTS, Bioengineering), in which three angles over time were plotted in the sagittal plane motion, namely the hip angle (absolute), the knee angle (relative), and the ankle angle (relative). The hip angle is formed by the thigh axis and the vertical of the hip passing through the pelvis. The knee angle is formed by the extension of the thigh axis and shank axis, whereas the ankle angle is formed by the shank axis and foot axis, passing over the lateral malleolus and the fifth metatarsal segment. Two gait cycles for each leg were sampled; the average was calculated to plot two angle–angle and angle–time relations. The selected cycles were as close to the middle of the gait test duration as possible so that the participants could adapt themselves to the preselected speed on the treadmill.

The angle–angle diagrams were determined according to Cavanagh [30]. Five points were chosen (Figure 1B,C): toe off (TO), maximum knee flexion (MKF), maximum hip flexion (MHF), foot support (FS), and maximum flexion of the knee during the contact phase (MKF-C). The frame with the minimum value of the knee angle was selected for MKF, the frame with the minimum value of the hip angle was selected for MHF, and the frame with the minimum value of the knee angle during the contact phase was selected for MKF-C. For TO and FS, the considered frames corresponded to those in which the value of the X vector of velocity changed from negative to positive and vice versa [42]. The areas and perimeters of the angle–angle diagrams were calculated by using AutoCAD 2023 (Autodesk, San Rafael, CA, USA). The shape of the loops formed by the angle–angle diagrams was calculated by dividing the perimeter multiplied by the root mean square of the area ($\frac{Perimeter}{\sqrt{Area}}$) [32]. For each subject, the angular variation as a function of time in the three joints of the lower limb were also analysed (Figure 2).

Similarly, the angle–angle diagrams for two consecutive steps were averaged for each limb. The acquisition began with the TO corresponding to the same step used in the angle–angle analysis and ended with the following TO. The kinematic data were synchronised with the sEMG activity. The sEMG detection technique considered a full wave based on the root mean square (RMS) of two successive stance phases (Figure 2).

### 2.4. Postural Control and Synchronised sEMG

Each participant assumed an upright standing position on the force platform (Muscle-Lab, Ergotest Innovation AS) facing 1.5 m away from the walls, which were draped with white sheets; a 1 cm red dot was placed in front of the participant at eye level (Figure 3A). When necessary, the participant was allowed to wear spectacles to ensure they had normal binocular vision and could comfortably fixate on the red dot on the wall. The stance width, or intermalleolar distance (IMD), was defined as the distance between the medial malleoli (about 1.5 cm). The appropriate toe and heel positions were marked on the force plate to ensure a consistent position among trials for each participant [43]. For each trial, the body sway was recorded for 30 s while the participant stood as still as possible with their hands held relaxed laterally along their hips. Data from the force plate were collected with a sampling frequency of 100 Hz. Postural sway was measured by quantifying the displacement of the centre of pressure (COP), which represents the projection of the centre of mass on the platform. The forward–backward COP (COP_FB_) and right–left centre of pressure (COP_RL_) displacements were considered during offline analysis (Figure 3B–D) [44]. Spatial resolution of 0.1–0.2 mm was assessed in the COP_RL_ and COP_FB_ directions [43]. Body sway while standing upright was assessed in the bipodalic static (BPS) and bipodalic dynamic (BPD) conditions [45] with open eyes (OE) or closed eyes (CE). The four trials—BPS-OE, BPS-CE, BPD-OE, and BPD-CE—were performed in a random order. The BPD conditions are represented by an external stimulus generated by a pendulum system (Figure 3A). The pendulum, placed behind the subject, impacted the dorsal part corresponding to the area between the inferior angles of the two scapulae, generating a slight perturbation. The stimulus was produced three times every 7 s (at 7, 14, and 21 s) during the 30 s BPD-OE and BPD-CE trials. During the OE condition, the participant kept their gaze fixed on the red dot. In the BPS conditions, the path displacements of the COP (in millimetres) were analysed during the entire 30 s trial. In the BPD conditions, three 1.5 s windows were analysed; each of them began when the ground reaction force changed due to the external stimulus (Figure 3C,D). sEMG was recorded synchronously with the body sway. For the BPS trials, the signal was recorded for the entire 30 s duration. For the BPD trials, the signal was analysed in 0.5 s windows around the sEMG peak response due to external perturbation (Figure 3E). The sEMG values obtained in the three windows were considered for analysis in each muscle.

### 2.5. Statistical Analysis

The XLSTAT 2013.2.07 software (Addinsoft; New York, NY, USA) was used for statistical analysis. The Shapiro–Wilk W test revealed that the distribution of the data deviated from a normal Gaussian distribution; therefore, non-parametric statistical procedures were used. The angle–time profiles were compared with the Kolmogorov–Smirnov test. The other dependent variables were compared with the Friedman test for each group during repeated measures over time by a Wilcoxon test for within-group comparisons and by a Mann–Whitney test for between-group comparisons to determine differences. The Bonferroni correction was used to adjust the *p* value in relation to the number of contrasts carried out. Statistical significance was set at α = 0.05, and the effect size was determined using Hedges’ g, which was considered to be small at g < 0.5, moderate at 0.5 < g < 0.8, and large at g > 0.8 [46].

The discriminant behaviour of the significant variables with respect to the ASD neuromuscular profile was used to estimate their potential diagnostic performance for ASD assessment. Given the available sample size, non-parametric receiver operating characteristic (ROC) curve analyses were performed using the STATA software (version 17; StataCorp LLC, College, Station, TX, USA). For each response, the AUC—an indicator of the sensitivity and specificity of a test—was estimated. The bootstrap standard errors and 95% confidence interval were calculated. The AUC ranges from 0.5 (no discrimination) to 1.0 (perfectly accurate test); 0.7–0.8 is considered acceptable, 0.8–0.9 is considered excellent, and >0.9 is considered outstanding [47]. Furthermore, given the best diagnostic performance of the outcome knee–angle diagram, the optimal diagnostic cut-off point was determined according to the Youden criteria.

## 3. Results

The differences between ASD and TD were not statistically significant in any of the anthropometric measured variables (*p* > 0.05).

The angle–time profiles determined at the hip, knee, and ankle joints did not show significant differences between the right and left leg within each group (ASD or TD; *p* > 0.05). Moreover, there were no significant angle–time profile differences between the ASD and TD groups (*p* > 0.05) (Figure 4).

The area, perimeter, and shape of the hip–knee and knee–ankle diagrams did not show significant differences between the right and left legs within each group (ASD or TD; *p* > 0.05). On the contrary, the area described by the knee–ankle diagrams for the right leg showed a significant difference between the ASD and TD groups (*p* < 0.05; ES = 2.32). However, the shape and perimeter defined by the knee–ankle diagram were not significantly different between the groups (*p* > 0.05). For the left leg, there were no significant differences between the ASD and TD groups (*p* > 0.05; Figure 5). The hip–knee diagrams showed no significant differences in the area, perimeter, and shape between the ASD and TD groups (*p* > 0.05; Figure 5).

During the contact phase of gait, the sEMG activity of the lateral gastrocnemius (LG) was significantly higher in the TD group compared with the ASD group (*p* < 0.05; ES = 1.39), whereas the other muscles showed similar activation in both groups (*p* > 0.05) (Figure 6).

The differences in body sway between the ASD and TD groups, analysed by quantifying COP_RL_ and COP_FB_ displacements in the BPS-OE, BPS-CE, BPD-OE, and BPD-CE conditions, were not significant (*p* > 0.05) (Appendix A). The right-to-left and forward–backward sway paths for the BPD conditions showed significant decreases in each group: the ASD group showed a decrease in the sway path for the BPD-OE condition (*p* < 0.05; ES = 0.96; ES = 1.14) while the TD group exhibited a decrease in the sway path for the BPD-CE condition (*p* < 0.05; ES = 1.19; ES = 1.07) (Appendix A).

The sEMG activity recorded for the BPS-CE and BPS-OE conditions revealed significantly higher activation of the LG in the TD group compared with the ASD group (*p* < 0.05; ES = 0.83; ES = 0.88). The activation of the other muscles was similar between the two groups (*p* > 0.05) (Appendix A). In the BPD conditions, the sEMG responses of the leg muscles were not different between the ASD and TD groups (*p* > 0.05; Appendix A). However, in individuals with TD, the TA showed significant responses during the BPD-OE and BPD-CE conditions (*p* < 0.05; ES = 0.70; ES = 0.68), whereas in individuals with ASD, the LG was involved during the BPD-OE condition (*p* < 0.05; ES = 0.46), and the TA was involved during the BPD-CE condition (*p* < 0.05; ES = 0.21) (Appendix A).

The area under the ROC curves (AUCs) for the four significant variables as predictors of ASD were: 0.84 (95% CI 0.656–0.1.00; *p* = 0.0001) for sEMG activity of the LG during balance in the BPS-CE condition, 0.84 (95% CI 0.659–1.00; *p* = 0.0001) for sEMG activity of the LG during balance in the BPS-OE condition, 0.81 (95% CI 0.603–1.00; *p* = 0.0001) for sEMG activity of the LG during walking, and 0.94 (95% CI 0.850–1.00; *p* = 0.0001) for the knee–ankle diagram during walking (Figure 7A–D).

## 4. Discussion

The purpose of this investigation was to characterise locomotion and postural control in young adults with ASD and matched TD controls during walking at a standardised speed and while standing upright. We also synchronously recorded the sEMG activity of leg muscles for both motor tasks. Our two main findings support our hypothesis: first, young adults with ASD exhibited a reduced area in the knee–ankle diagram compared with young adults with TD; second, individuals with ASD showed lower LG activation in the stance phase during walking and postural control while standing upright compared with individuals with TD.

### 4.1. Walking at Standardised Speed

Our kinematic analysis of walking in the present study is not comparable with other published studies [9,12,14,15,22,24,28,48,49] due to marked differences in experimental designs, participants, and descriptor measures of gait. Specifically, in the mentioned studies, the age of participants ranged from 6 to 14 years, except for the studies by Weiss et al. [9] and Armitano et al. [28], although in both studies the authors analysed spatial and temporal measures of gait. Moreover, Weiss et al. [9] selected young adults diagnosed with ‘severe autism’ and ‘verbal communication disorders’. Some of our findings are similar to those of Hallett et al. [23] regarding the ankle ROM, which was significantly lower in individuals with high-functioning autism than the neurotypical controls. In the present study, the knee and ankle ROM tended to be different during a gait cycle between the ASD and TD groups, although the difference was not significant.

Our study is the first that has analysed the gait of individuals with ASD by using angle–angle diagrams. The loop described by the knee–ankle diagram in individuals with ASD compared with individuals with TD revealed a reduced area and a trend to show a different shape in the right leg, whereas the perimeter tended to be reduced in the left leg. The area is a function of the knee and ankle angles and is the expression of the total conjoint angular ROM performed by the knee and ankle joints during a complete gait cycle [32]. Therefore, a smaller area implies that individuals with ASD produce fewer simultaneous rotations at the knee and ankle joints than individuals with TD. However, a knee–ankle diagram that shows ROM changes for one angle and no change for the other angle could alter the perimeter but not the area [32]. In other words, the perimeter between the knee and ankle in the left leg of individuals with ASD provides an indication of uncoordinated conjoint ROM. Although the absolute value of the shape cannot be associated with a specific geometrical shape, it has potential value to underline the changes in shape described by an angle–angle diagram. For example, when the ROM remains constant but the movements are not coordinated, the value that quantifies the shape tends to increase [32]. In summary, the area, perimeter, and shape of a closed loop can be interpreted quantitatively from angle–angle diagrams. These factors highlight differences between individuals with ASD and individuals with TD in temporal and angular data that are linked simultaneously with the knee and ankle joints.

### 4.2. Postural Control

We assessed postural control while standing upright by measuring the sway path of the COP in four conditions (BPS-OE, BPS-CE, BPD-OE, and BPD-CE). The right-to-left sway path in the BPS conditions tended to be lower in individuals with ASD than in individuals with TD, particularly when their eyes were closed (*p* = 0.063). Post hoc analysis of our data showed that the present study was underpowered for the latter variable with a power (1 − β) equal to 0.38. A larger number of participants (n = 72) would have increased the power to 0.90 to detect a significant difference. When comparing our results with published findings, we must consider the age of the participants because altered postural control in individuals with ASD is most evident in children before the critical period of sensorimotor development [17,50,51,52,53]. Similarly to our study, other investigations did not find significant differences in postural control between individuals with ASD and individuals with TD [20,21,53]. On the contrary, during experimental conditions that require sensory integration, adults with ASD show a larger right-to-left sway path than individuals with TD when a visual stimulus is displayed and attentional demands are associated with postural control [54], and a greater forward–backward sway path speed than individuals with TD in destabilising sensory stimuli (proprioception and vision) [55]. Additionally, in the latter study, when exposed to a prolonged challenging sensory condition over time (3 min), adults with ASD exhibited a reduced adaptative ability (sway path speed), highlighting an inflexible postural pattern [55].

In the present study, we used a novel paradigm to assess the role of sensory integration in postural behaviour. We dynamically altered proprioception over time by inducing repeated external stimuli (Figure 3A). The adaptive responses to the repeated perturbations showed significant differences within each group. Specifically, individuals with ASD were hyperreactive in the right-to-left sway with their eyes open and closed, whereas they only showed hyperreactivity in the forward–backward sway when their eyes were open. On the other hand, individuals with TD exhibited hyperreactive responses for the right-to-left and forward–backward sways only when their eyes were closed. These findings are consistent with the results reported by Doumas et al. [20], despite the different experimental conditions between their study and ours.

The hyperreactivity that we found in young adults with ASD suggests an association between environmental perception and the subsequent postural behaviour. Specifically, the sensitivity of their postural control system increases with a gradual increase in sensory integration demands, reflecting an impairment of a sensory channel-specific (i.e., vision). This explanation is based on the results obtained during BPD conditions: compared with individuals with TD, individuals with ASD showed different adaptive responses in forward–backward sway relative to the presence of visual processing. In this regard, the vision of individuals with ASD negatively affected their postural behaviour (denoted by a hyperreactive response) during forward–backward sway but not during right-to-left sway. A possible explanation could be related to a greater reliance on vision during movement in the forward–backward direction (in individuals with ASD, this sensory channel is difficult to integrate) than the right-to-left direction because vision becomes less effective in detecting lateral motion [56,57]. In other words, the forward–backward and right-to-left sway components are regulated with independent strategies, and the right-to-left direction is more complex in terms of control as reflected in modulation of electroencephalographic patterns [58], and the exclusion of the visual channel to reduce the demands of sensory integration is not the sufficient condition to avoid a hyperreactive response in individuals with ASD. Therefore, we cannot exclude that in individuals with ASD, the hyperreactivity of the postural control in the right-to-left direction may be influenced by the altered activity in the cerebral cortex [4].

The cerebral cortex is a primary site of brain dysfunction in individuals with ASD. It presents bilateral minicolumn abnormalities (minicolumns are composed of radially orientated arrays of pyramidal neurones [layers II–VI], interneurones [layers I–VI], axons, and dendrites) in cortical areas 3, 4, 9, 17, 21, and 22 [4]. The minicolumns have been hypothesised to be the smallest radial unit of information processing in the cortex. Hence, there seems to be a disturbance in cortical synchronisation and inhibition in individuals with ASD [4]. These aspects underline that postural instability is the result of reduced capacity for multimodal sensory integration (visual, vestibular, and proprioception) and provide support for the connectivity model of how ASD affects the brain, with information integration as a common denominator [4].

### 4.3. Mechanical and Neuromuscular Interactions

Regarding the postural control strategies, during the BPD conditions, the two groups showed the same neuromuscular strategy when the tasks were performed without visual processing (closed eyes). The TA showed reduced activity after the first perturbations, while the VL, BF, and LG maintained approximately constant activities. Nevertheless, individuals with ASD showed the highest postural perturbation during the BPD-OE condition, whereas the individuals with TD showed the greatest perturbation during the BPD-CE condition.

When the tasks were performed with visual processing, the two groups showed an inverted strategy concerning activation of the TA and LG. In the individuals with TD, after the first perturbation, the TA activity decreased, while the VL, BF, and LG activity remained constant. The neuromuscular strategy used by the individuals with TD did not change depending on whether the eyes were open or closed, but they were not hyperreactive when their eyes were open. Hence, whether the eyes are open does not influence the neuromuscular strategy, but it can alter the sensitivity of the neural processes. On the contrary, the individuals with ASD showed a decrease in LG activation during the BPD conditions, while VL, BF, and LG activity tended to be constant. Differently from the individuals with TD, the individuals with ASD showed an altered neuromuscular strategy for the TA and LG for the BPD-OE and BPD-CE conditions. This altered strategy adopted by the individuals with ASD did not control their hyperreactivity and sway when their eyes were open or closed, particularly in the right-to-left direction.

The postural control theory indicates that the body sway is regulated using two different control strategies: the ankle strategy predominantly regulates sway in the forward–backward direction, whereas the hip strategy regulates sway in the right-to-left direction [34]. The ankle strategy re-establishes the equilibrium of the body by producing torques around the ankle joints; the neuromuscular pattern primarily involves the ankle joint muscles, and then it extends towards the thigh and trunk muscles. On the contrary, in the hip strategy, the neuromuscular activation is proximal to distal to produce compensatory shear forces at the hip joint and to support ankle torque. Healthy subjects exposed to perturbations, older individuals, or people affected by pathology can present complex combinations of the pure ankle and hip strategies by activating muscles of the ankle and hip or by increasing co-contraction [34,57].

Our results showed that in individuals with ASD, when visual processing is associated with an unexpected perturbation (i.e., the BPD-OE condition), it inverts the neuromuscular pattern of the TA and LG within the ankle strategy. On the other hand, in individuals with TD, the visual processing does not modify the ankle strategy; rather, the amplitude of muscle activation was increased. On the contrary, during the BPS conditions, visual processing does not alter the strategy for either group. The only difference between the two groups for the BPS conditions was lower LG activation for the individuals with ASD when their eyes were closed. Moreover, individuals with ASD showed lower activation of the LG during the stance phase of walking compared with individuals with TD. Taken together, these results highlight an alteration in the ankle strategy during postural control and walking in individuals with ASD.

### 4.4. Predictive Ability of the Measured Variables

We analysed differences in several variables between the ASD and TD groups based on the AUC, an effective way to summarise the overall diagnostic accuracy of a test. The AUCs relative to the sEMG activity of the LG during walking and during postural control (BPS-OE and BPS-CE) showed an excellent ability to diagnose individuals with ASD (the values ranged from 0.8 to 0.9). There was an outstanding ability to diagnose individuals with ASD based on the knee–ankle diagram determined during walking: the AUC of 0.94 indicates a 94% chance, with *p* < 0.0001, that the knee–ankle diagram will correctly distinguish an individual with TD from an individual with ASD based on the dimension (area) of the loop. The sensitivity (ability to detect the disease when it is truly present) and the specificity (the probability of excluding the disease state when it is not present) were 0.8 (95% CI 0.55–1.04) and 1.0 (95% CI 0.81–1.18), respectively. These values are promising for considering the knee–ankle diagram a desirable test either in a diagnostic setting (higher sensitivity is often considered desirable) or a screening setup (higher specificity is desirable) [47].

### 4.5. Limitations

The present study has some limitations. First, the study was underpowered for the secondary functional outcomes (angle–time relations and body sway). Post hoc analysis demonstrated that with a larger number of participants (n = 70–80), the chance of detecting significant differences in other variables would have increased to 90%. However, the relatively small sample size pointed out that knee–ankle diagram and the sEMG activity of LG were the more sensitive parameters to discriminate between the ASD and TD groups. Second, we did not normalise the sEMG activity and recorded it in the muscles of one leg (the dominant leg). Recording the activity in both leg muscles would have allowed us to better describe the pattern of activation and the interactions between the muscles of the two legs during the gait cycle and postural control. Third, the participants involved were young male adults with ASD; therefore, the results are only generalisable to individuals with similar characteristics to those that took part in the present study (e.g., sex, age, IQ, etc.).

## 5. Conclusions

This study demonstrates the potential utility of the knee–ankle diagram as a sensitive and specific movement descriptor to differentiate between individuals with ASD and individuals with TD and to assess the effectiveness of exercise and/or psychological interventions in clinical or longitudinal studies. Additionally, synchronised sEMG with quantitative measurements of body sway while standing upright with altered sensory information (dynamic) had better sensitivity than normal quiet standing (static) to characterise the inverted neuromuscular activation of the LG and TA adopted by individuals with ASD during postural control. In summary, the neuromuscular activation of the TA and LG is the common denominator of an altered ankle strategy during locomotion and postural control while standing upright in individuals with ASD.

## Figures and Tables

**Figure 1 jfmk-09-00185-f001:**
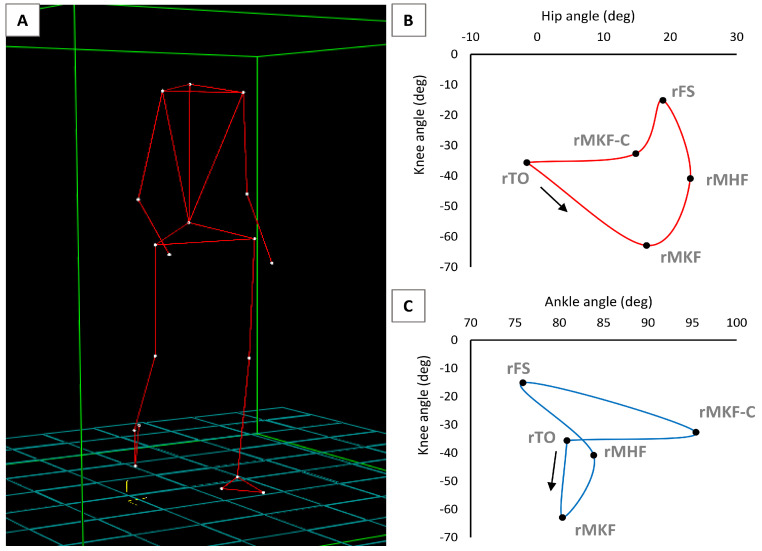
Determination of an angle–angle diagram of a person with ASD. (**A**) Body reconstruction obtained by the SMART Motion Capture System (BTS, Bioengineering, Milano). (**B**) Hip angle–knee angle diagram of the right leg of a person with ASD (rTO = right toe off, rMKF = maximum knee flexion, rMHF = maximum hip flexion, rFS = right foot strike, rMKF-C = right maximum knee flexion during the contact phase). (**C**) Knee angle–ankle angle diagram of the right leg of a person with ASD.

**Figure 2 jfmk-09-00185-f002:**
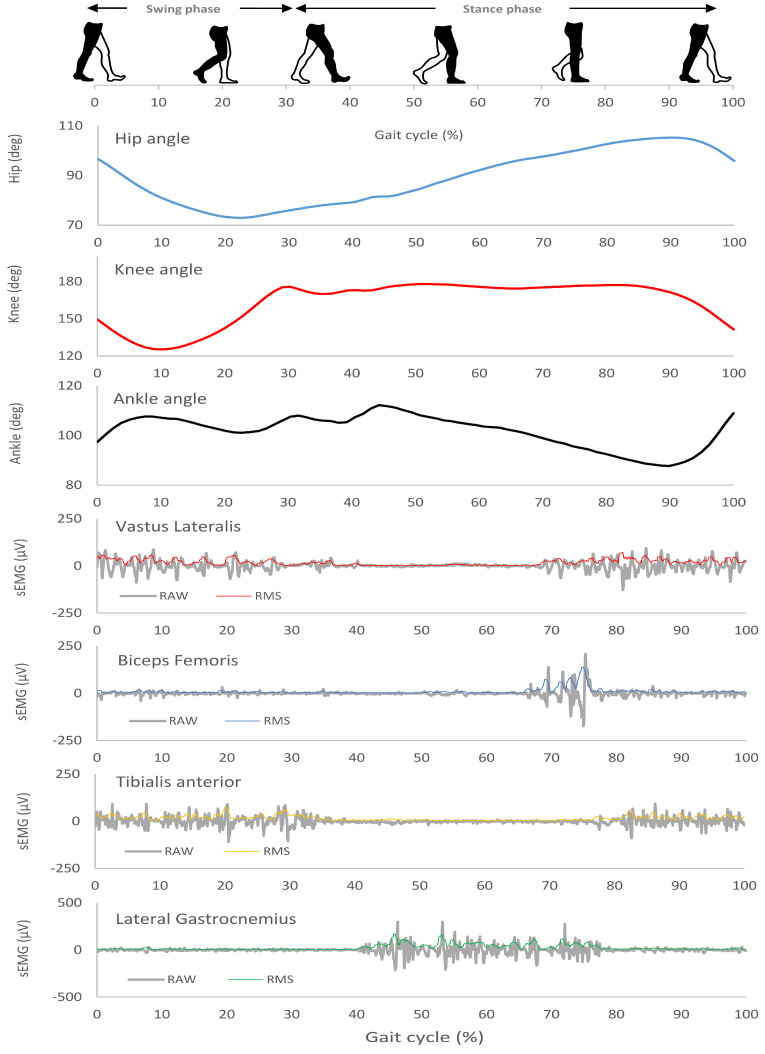
Representative example of a person with ASD during a gait cycle. The angle–time variations of the lower limb and sEMG of the leg muscles.

**Figure 3 jfmk-09-00185-f003:**
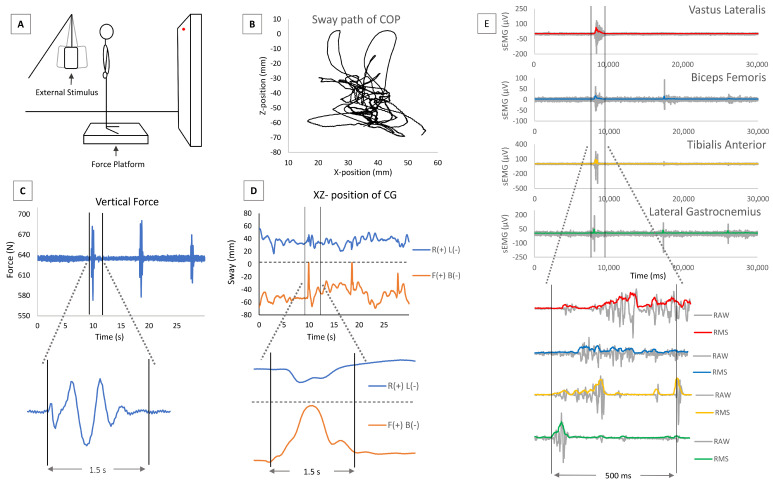
(**A**) Experimental setup for assessment of body sway in dynamic upright standing conditions; (**B**) body sway path of COP; (**C**) vertical ground reaction force recording during body sway in dynamic conditions; (**D**) COP displacement in forward–backward (F(+) B(−)) and right–left (R(+) L(−)) components, during the dynamic postural assessment; (**E**) sEMG activity during postural assessment. (**C**–**E**) examples of the windows to analyse the signal during dynamic conditions.

**Figure 4 jfmk-09-00185-f004:**
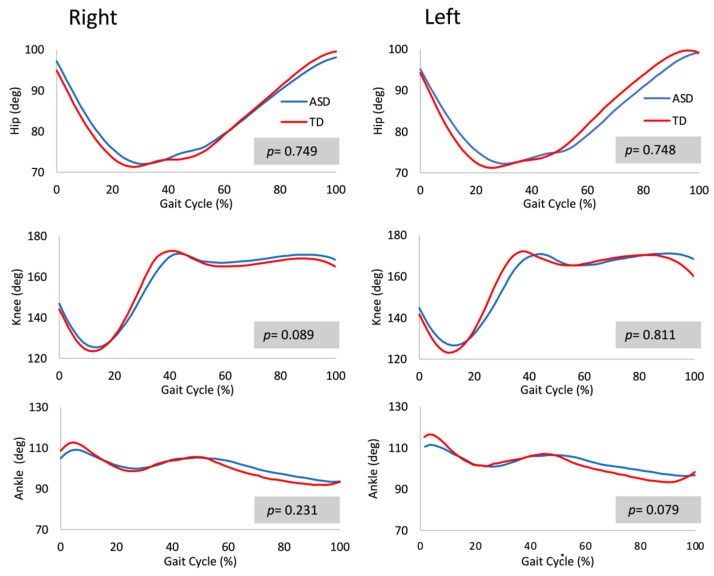
Angle–time profiles (hip, knee, and ankle) during walking on a treadmill (speed = 3 km/h) in the experimental (ASD group) and typical developmental (TD group) subjects.

**Figure 5 jfmk-09-00185-f005:**
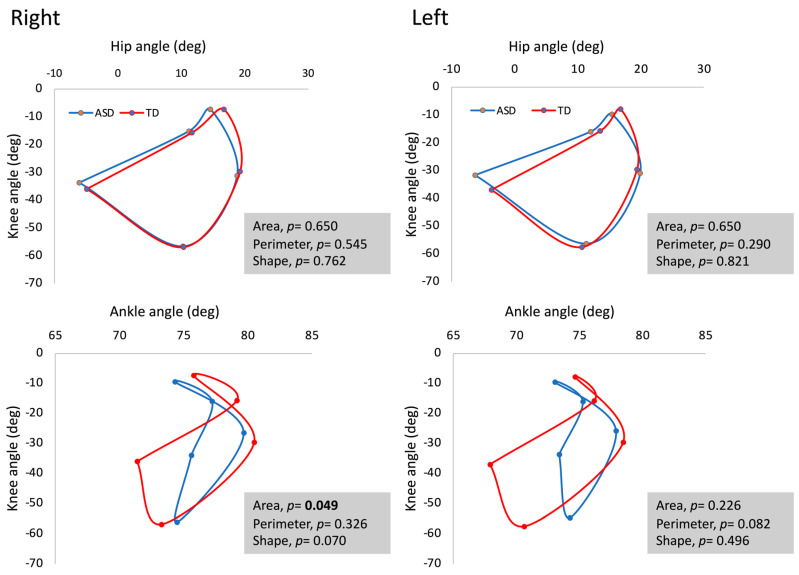
Angle–angle diagrams (hip–knee and knee–ankle) during walking on a treadmill (speed = 3 km/h) in the experimental (ASD group) and typical developmental (TD group) subjects.

**Figure 6 jfmk-09-00185-f006:**
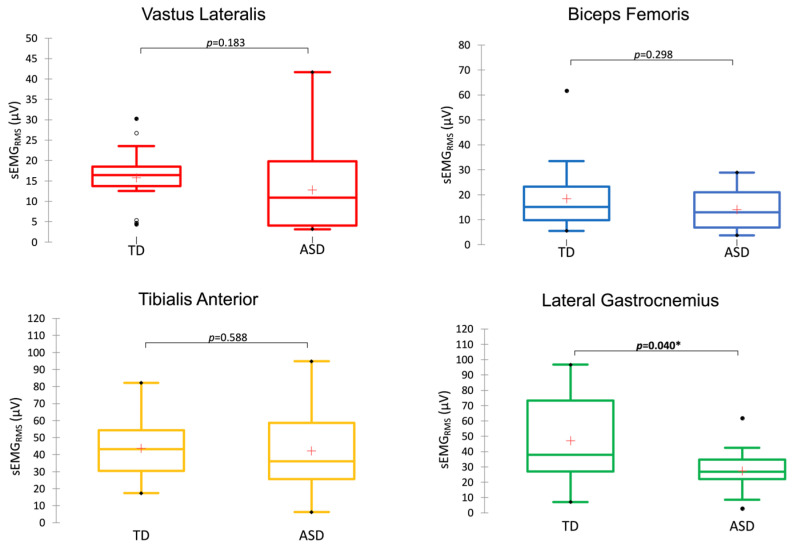
sEMG activity in the stance phase of two consecutive steps in the experimental (ASD group) and typical developmental (TD group) subjects. **p* < 0.05.

**Figure 7 jfmk-09-00185-f007:**
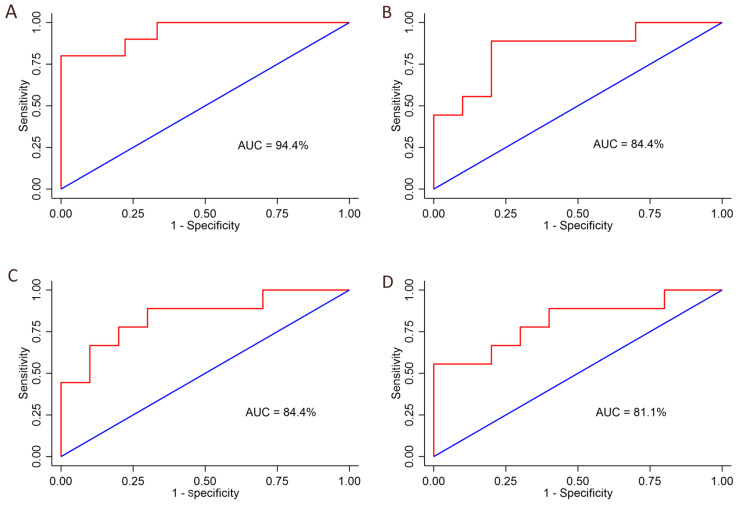
ROC curve of: (**A**) knee–ankle diagram during walking; (**B**) sEMG_RMS_ of lateral gastrocnemius (LG) during bi-podalic static (BPS) open eyes (OE) condition; (**C**) sEMG_RMS_ of LG during BPS closed eyes (CE) condition; (**D**) sEMG_RMS_ of LG during walking. ROC, receiver operating characteristic curve; AUC, area under curve.

## Data Availability

The raw data supporting the present study will be made available by the authors without restriction.

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
