# Peer review of "Locomotion and Postural Control in Young Adults with Autism Spectrum Disorders: A Novel Kinesiological Assessment"

_jfmk, 2024, doi:10.3390/jfmk9040185_

Round 1

Reviewer 1 Report

Comments and Suggestions for Authors

The purpose of this study was to assess gait using a novel approach that plotted two adjacent joint angles and postural control in individuals with autism (ASD) and individuals with typical neurodevelopmental (TD). A total of 20 young men completed the study (10 ASD, 10 TD) and walked for 30 s at 3 km/h on a treadmill while simultaneous kinematics and EMG activity measurements were undertaken. The main findings were that differences existed between ASD and TD for the knee angle diagram, EMG activity in the LG muscle during the gait contact phase, and EMG activity in the LG and TA muscles in postural conditions. The authors concluded that their knee–ankle diagram was a sensitive and specific movement descriptor to differentiate ASD from TD individuals. Finally, it was concluded the reduced LG activation was likely responsible for the reduced area in the knee–ankle diagram and ‘toe-walking’ in ASD and therefore represents the common denominator of an altered ankle strategy during locomotion and postural control.  

I believe the study had the severall strengths: 1) overall well-written and easy to understand; 2) novel methodology; 3) the figures were well-done; 4) ASD is an important research topic in general due to the prevalence and lower limb control has been shown to be impaired in many studies; (5) most of the limitations of the study were acknowledged (but see below). Overall, I think the article adds to the ASD and gait literature and id appropriate for JFMK. It should be of interest to researchers in ASD and biomechanics and several adjacent fields. 

I don’t think the article has any major weaknesses or issues. Accordingly, I have only one major comment and several minor comments the authors should address. 

Major:

1. Although most of the EMG methodology and processing were very good. The one issue I have is that it appears that the authors used RMS for the EMG amplitude, which is done in a lot of studies however. I know it is more difficult and time consuming, but I feel it would have been better to perform manual muscle tests for the muscles studied and normalize the EMG during the tasks to the maximal EMG during the manual muscle tests. Perhaps this should be added as another limitation in the Limitations section? Or the authors can address as to why this is not a major issue. 

Minor:

1. Could walking on a treadmill give different results than walking overground? There are some differences in some studies between the two. Could there be differences in a population like this especially if they are not familiar with treadmill walking. Should this issue be addressed in somewhere the paper or in the limitations section?

2. Although overall well-written there are a few typographical and formatting errors. I point out a few examples below but more proofreading is needed overall.

A. No space before the bracket many times in the in text citations for example lines 35, 41, 74. Also I think too many spaces before the citation in line 67.

B. Line 161 in text citation and afterward I think the underline of the period is a track changes that was not removed.

C. Line 15, I don’t think “The” is needed

2. Line 440 is it true that cerebral cortex is considered the primary site, I thought perhaps cerebellum is considered the origin of more of the dysfunctions, but I could be wrong. Does the reference provided say that?

3. May be a few consistencies in the bibliography such as the title formatting of reference 12 and 42 relative to all the other references.

Comments on the Quality of English Language

very minor proofreading needed

Author Response

General comment: The purpose of this study was to assess gait using a novel approach that plotted two adjacent joint angles and postural control in individuals with autism (ASD) and individuals with typical neurodevelopmental (TD). A total of 20 young men completed the study (10 ASD, 10 TD) and walked for 30 s at 3 km/h on a treadmill while simultaneous kinematics and EMG activity measurements were undertaken. The main findings were that differences existed between ASD and TD for the knee angle diagram, EMG activity in the LG muscle during the gait contact phase, and EMG activity in the LG and TA muscles in postural conditions. The authors concluded that their knee–ankle diagram was a sensitive and specific movement descriptor to differentiate ASD from TD individuals. Finally, it was concluded the reduced LG activation was likely responsible for the reduced area in the knee–ankle diagram and ‘toe-walking’ in ASD and therefore represents the common denominator of an altered ankle strategy during locomotion and postural control.   

I believe the study had the several strengths: 1) overall well-written and easy to understand; 2) novel methodology; 3) the figures were well-done; 4) ASD is an important research topic in general due to the prevalence and lower limb control has been shown to be impaired in many studies; (5) most of the limitations of the study were acknowledged (but see below). Overall, I think the article adds to the ASD and gait literature and id appropriate for JFMK. It should be of interest to researchers in ASD and biomechanics and several adjacent fields.  

I don’t think the article has any major weaknesses or issues. Accordingly, I have only one major comment and several minor comments the authors should address.  

Response to general comment: First, we would like to thank Editor and Reviewers for their comprehensive review of our paper and for their comments. Your comments helped us to rethink our study and paper. It was very useful. As you can see, we accepted many of your suggestion. However, in some cases we tried to give answers to your questions and to reason of our opinion. Thank you very much your kind assistance in advance.   

Red letter shows our responses to your comments. Changes and corrections highlighted with yellow were made in the text where red letter indicate the corrections. 

Major comment: 

Comment1: Although most of the EMG methodology and processing were very good. The one issue I have is that it appears that the authors used RMS for the EMG amplitude, which is done in a lot of studies however. I know it is more difficult and time consuming, but I feel it would have been better to perform manual muscle tests for the muscles studied and normalize the EMG during the tasks to the maximal EMG during the manual muscle tests. Perhaps this should be added as another limitation in the Limitations section? Or the authors can address as to why this is not a major issue.  

Response 1: We agree with the reviewer that normalization is the best procedure to standardize the EMG amplitude in many tasks. We have included this issue in the limitation section. Anyway, in the literature it has been reported that the dynamic propriety of EMG pattern, during gait, are still maintained (Ricamato AL, Hidler JM. Quantification of the dynamic properties of EMG patterns during gait. J Electromyogr Kinesiol. 2005;15(4):384-392. doi:10.1016/j.jelekin.2004.10.003; Cronin NJ, et al. Spatial variability of muscle activity during human walking: the effects of different EMG normalization approaches. Neuroscience. 2015;300:19-28. doi:10.1016/j.neuroscience.2015.05.003). In addition, it has been reported that there are no inter-participant statistical differences during the stance phase of gait between values normalized and non-normalized (Ghazwan A, et al. Can activities of daily living contribute to EMG normalization for gait analysis?. PLoS One. 2017;12(4):e0174670. doi:10.1371/journal.pone.0174670).  

Minor comments: 

Comment 1: Could walking on a treadmill give different results than walking overground? There are some differences in some studies between the two. Could there be differences in a population like this especially if they are not familiar with treadmill walking. Should this issue be addressed in somewhere the paper or in the limitations section? 

Response 1: In a recent review, slight differences have been reported and they regarding the hip joint (Semaan MB, et al. 2022. Is treadmill walking biomechanically comparable to overground walking? A systematic review. Gait Posture. doi: 10.1016/j.gaitpost). However, in our study the hip joint does not show alterations during gait in ASD compared with typical development. We believe that the use of the treadmill was a strength of the study, rather than a limitation, because the kinematic variables, that are speed dependent, were standardized to the same speed for all subjects. We also emphasize, as reported in the text (lines 159-164), that all participants familiarized with treadmill walking.   

Comment 2: Although overall well-written there are a few typographical and formatting errors. I point out a few examples below, but more proofreading is needed overall. 

Comment 2A: No space before the bracket many times in the citations for example lines 35, 41, 74. Also I think there are too many spaces before the citation in line 67. 

Response 2A: We have checked the text and corrected the missing brackets or excess spaces. 

Comment 2B: Line 161 in text citation and afterward I think the underline of the period is a track changes that was not removed. 

Response 2B: We have corrected in the text. 

Comment 2C: Line 15, I don’t think “The” is needed 

Response 2C: We have corrected in the text, removing “the”. 

Comment 3: Line 440 is it true that cerebral cortex is considered the primary site, I thought perhaps cerebellum is considered the origin of more of the dysfunctions, but I could be wrong. Does the reference provided say that? 

Response 3: Cerebellar alterations seem to be one of the causes of motor impairments in this population, as we reported in the introduction. However, cerebral cortex alterations (as reported by citation 4), considering their function of multisensory integration and programming in the execution of movement are considered one of the primary sites of ASD dysfunction. 

Comment 4: May be a few consistencies in the bibliography such as the title formatting of reference 12 and 42 relative to all the other references. 

Response 4: We have reformatted the title of the two references. 

Reviewer 2 Report

Comments and Suggestions for Authors

I would like to thank you very much for the opportunity to review the manuscript entitled Locomotion and Postural Control in Young Adults with Autism Spectrum Disorders: A Novel Kinesiological Assessment. First of all, I would like to congratulate the authors on the enormous amount of work they put into preparing this manuscript. The title and purpose are formulated correctly. The introduction comprehensively explains the background of the research problem, which was the assessment of locomotion and postural control in people with ASD. The references are well selected, taking into account the latest publications on the subject. Older bibliographic entries are used where necessary. The methodology is described clearly. The presentation of the results in graphical form deserves special attention, which makes it easier for the reader to understand them.

I have only a few minor comments:

- I suggest moving the specific atropometric results of the participants (lines 133-13) from the methods section to the results section

- Please explain on what basis such a high effect size was established when calculating the sample size?

- Due to the lack of normal distribution and the use of nonparametric tests, I suggest presenting the results as medians with ranges, not means with standard deviations

Author Response

General comment: I would like to thank you very much for the opportunity to review the manuscript entitled Locomotion and Postural Control in Young Adults with Autism Spectrum Disorders: A Novel Kinesiological Assessment. First of all, I would like to congratulate the authors on the enormous amount of work they put into preparing this manuscript. The title and purpose are formulated correctly. The introduction comprehensively explains the background of the research problem, which was the assessment of locomotion and postural control in people with ASD. The references are well selected, taking into account the latest publications on the subject. Older bibliographic entries are used where necessary. The methodology is described clearly. The presentation of the results in graphical form deserves special attention, which makes it easier for the reader to understand them. 

Response to general comment: First, we would like to thank Editor and Reviewers for their comprehensive review of our paper and for their comments. Your comments helped us to rethink our study and paper. It was very useful. As you can see, we accepted many of your suggestions. However, in some cases we tried to give answers to your questions and to reason of our opinion. Thank you very much your kind assistance in advance.   

Red letters show our responses to your comments. Changes and corrections highlighted with yellow were made in the text where red letter indicate the corrections.  

I have only a few minor comments: 

Comment 1: I suggest moving the specific anthropometric results of the participants (lines 133-13) from the methods section to the results section 

Response 2: Usually, the anthropometric characteristics are reported in the section subjects. We prefer to keep the original formatting.  

Comment 2: Please explain on what basis such a high effect size was established when calculating the sample size? 

Response 2: The effect size used to determine the sample size was relative to the angle-angle diagram (knee-ankle). We hypothesized that it would be the primary variable to show differences between the two groups. A previous our study (32-Di Giminiani et al. 2022) showed that the knee-ankle angle diagrams were very sensitive discriminating the gait differences between subjects with multiple sclerosis over healthy controls due to a reduced peripheral limb control. In the present study, ASD gait is characterized by toe walking. Therefore, we hypothesized that the knee-ankle diagram could be the most sensitive variable to discriminate differences between the two groups. 

Comment 3: Due to the lack of normal distribution and the use of nonparametric tests, I suggest presenting the results as medians with ranges, not means with standard deviations 

Response 3: Thanks for the suggestion, we have changed the presentation of results and replaced the means and standard deviations with medians and ranges. 

Reviewer 3 Report

Comments and Suggestions for Authors

Dear authors

The manuscript is interesting, but I must address the following concerns:

1. The introduction is too large. Please, be concise and make your point addressing the main features to the present work.

2. The sample size calculation must be further explained. From what variable did you extract the ES? Insert a reference. The a priori sample size was 1 or 2-tailed? Explain why.

3. The predictive value of the present study is, at least, questionable. The sample size calculation raises doubts, impairing the generalization. I strongly recommend caution to interpret the present data.

4. The ASD participants were all classified as level 1? Do you consider that this could change the outcome? If so, please, clarify and make a statement.

5. The conclusion is too large. Consider only the highlights of the present findings to restrict your conclusion.

Author Response

General comment: Dear authors The manuscript is interesting, but I must address the following concerns: 

Response to general comment: First, we would like to thank Editor and Reviewers for their comprehensive review of our paper and for their comments. Your comments helped us to rethink our study and paper. It was very useful. As you can see, we accepted many of your suggestion. However, in some cases we tried to give answers to your questions and to reason of our opinion. Thank you very much your kind assistance in advance.   

Red letter shows our responses to your comments. Changes and corrections highlighted with yellow were made in the text where red letter indicate the corrections.  

Comment 1: The introduction is too large. Please, be concise and make your point addressing the main features of the present work. 

Response 1: We agree with the reviewer that the introduction is large. Anyway, we think that the length is justified by the experimental approach in which several variables were measured. The logical flow in the introduction explained the rationale of the study and it should facilitate the reader to the comprehension of the paper.   

Comment 2: The sample size calculation must be further explained. From what variable did you extract the ES? Insert a reference. The priori sample size was 1 or 2-tailed? Explain why. 

Response 2: The effect size (Hedges’ g) used to determine the sample size was based on the angle-angle (knee-ankle), which we hypothesized would be the primary variable to show differences between the two groups. The ES was based on the results of our previous study (32-Di Giminiani et al. 2022), it was shown that the knee-ankle angle diagrams are very sensitive to discriminate the gait differences between subjects with multiple sclerosis over healthy controls. The a priori sample size is two-tailed, which is considered a more conservative approach because we didn’t know the direction of the effect size. 

Comment 3: The predictive value of the present study is, at least, questionable. The sample size calculation raises doubts, impairing the generalization. I strongly recommend caution to interpret the present data. 

Response 3: If sample could show a statistically significant difference with high power. The sample size should be considered adequate. In addition, considering the heterogeneity of pathology, recruiting large samples with the same physical, cognitive and diagnostic characteristics is not possible. In fact, studies with high sample size also have high heterogeneity among participants. Considering this, we believe, with caution as specified in the paper, that our results are generalizable to a population with similar characteristics to our experimental group. 

Comment 4: The ASD participants were all classified as level 1? Do you consider that this could change the outcome? If so, please, clarify and make a statement. 

Response 4: We have 8 ASD level 1 and 2 ASD level 2. We do not believe that this could significantly affect the results, as the ability to perform a motor task does not correlate with the intellectual level (Mosconi 2015). In particular, we ensured that participants with a level 2 understood the instructions to perform the motor task in order to include them in the sample. We are aware that the results cannot be generalized to the entire population with ASD. 

Comment 5: The conclusion is too large. Consider only the highlights of the present findings to restrict your conclusion. 

 Response 5: We agree with the reviewer. We have shortened the conclusions.